# Nanostructure Design and Catalytic Performance of Mo/ZnAl-LDH in Cationic Orchid X-BL Removal

**DOI:** 10.3390/ma11122390

**Published:** 2018-11-27

**Authors:** Yin Xu, Tingjiao Liu, Yang Li, Yun Liu, Fei Ge

**Affiliations:** Department of Environment, College of Environment and Resources, Xiangtan University, Xiangtan 411105, China; xtu_ltj2016@126.com (T.L.); xtu_ly2012@163.com (Y.L.); liuyunscut@163.com (Y.L.); gefei@xtu.edu.cn (F.G.)

**Keywords:** preparation method, Mo/ZnAl-LDH catalyst, organic wastewater, room conditions, wet catalytic oxidation

## Abstract

The nanostructure of ZnAl-layered double hydroxide (ZnAl-LDH) was designed to promote the catalytic performance of Mo-based ZnAl-LDH (Mo/ZnAl-LDH) catalysts, in a catalytic wet air oxidation (CWAO) process, under room temperature and pressure, in degradation of dye wastewater. Four most commonly used preparation methods, traditional precipitation (TP), hydrothermal synthesis (HS), sol-gel (SG), and urea co-precipitation (UC) were employed to prepare the ZnAl-LDH. The resulting Mo/ZnAl-LDH samples were contrasted through surface area, crystal structure, chemical state, and morphology. The degradation of cationic orchid X-BL, under room temperature and pressure, was developed to determine the catalytic activity of these Mo/ZnAl-LDH samples. The results showed that the nanostructure of ZnAl-LDH, prepared by HS, enhanced the adhesion of the catalytic active component, thus Mo/ZnAl-LDH had the highest catalytic activity of 84.2% color removal efficiency and 73.9% total organic carbon removal efficiency. Specific Mo species, such as Na_2_Mo_2_O_7_, Mo dispersion, and O^2−^ ions were proved to be related with catalytic performance. These findings preliminarily clarified that LDHs preparation methods make a difference in the performance of Mo/LDHs.

## 1. Introduction

Catalytic wet air oxidation (CWAO) is an effective method for organic wastewater treatment in the 21st century [1,2], which allows the use of less severe reaction conditions. In addition, it is suited to the decomposition of refractory pollutants, even, thereby, reducing the capital and operating costs [3]. The choice of the catalyst in the CWAO process is significant and, thus, seeking an effective, robust, and low-cost catalyst, has been a research hotspot [4,5,6]. Heterogeneous catalysts containing small amounts of molybdenum oxides, and clay-immobilized molybdenum, have attracted considerable attention because of their wide structural variety and specific properties. Our previous studies show that a Mo-Zn-Al-O catalyst has an excellent performance on the degradation of dye wastewater, under room temperature and pressure [7,8]. To promote the catalytic activity of Mo-Zn-Al-O catalyst, attention has been focused on the nanostructure design.

Generally, the catalytic performance of catalyst is determinedly influenced by their structure and morphology. During the preparation process, many factors have an impact on the final catalyst properties [9,10,11,12,13,14]. Specially, the nanostructure of the carrier, affect the metal dosage because of the interactions between the carrier and the metal precursor [15,16,17]. Layer double hydroxides (LDHs) have found an increasingly wide utilization in the catalyst carrier. Their main structure features are highly-related to their preparation method [18]. The most common synthesis method is the traditional precipitation (TP), at a constant pH. The urea co-precipitation (UC), sol-gel (SG), and hydrothermal synthesis (HS) create an alkaline environment and well-crystallized LDHs. Another popular method is the reconstruction from oxide generated by calcining LDHs, taking advantage of the “memory effect” also allows the intercalation of any desired anion, such as biomedical or active metal anions. Although the way to prepare LDHs was reported to have a significant effect on the final properties [19], the details of the relationship between the nanostructure of LDHs carrier, and the active metal precursor, are not fully understood.

It is reasonable to believe that the control of interaction between the active Mo component and LDHs carrier is a new direction for the newly-developed CWAO catalyst. Four typical methods were adopted for the ZnAl-LDH preparation, in the present study. The as-prepared samples were characterized by N_2_ adsorption-desorption, X-ray diffraction, X-ray photoelectron spectrum, and transmission electron microscopy. In brief, the cationic orchid X-BL was harmful to human health, under certain conditions, which was representative of dye wastewater [20]. Thus, their catalytic efficiency on the cationic orchid X-BL degradation was investigated, in detail. We optimized the formation to obtain a Mo/ZnAl-LDH catalyst with a high catalytic performance, on the basis of these results.

## 2. Materials and Methods

### 2.1. Materials

Cationic orchid X-BL was purchased from the Shanghai Luojing Dyeing Chemical Co., Ltd. (China). The chemicals used for this project, including Zn(NO_3_)_2_·6H_2_O (99%), Al(NO_3_)_3_·9H_2_O (99%), NaOH (99%), C_6_H_8_O_7_·H_2_O (99.5%), CH_4_N_2_O (99%), and (NH_4_)_6_Mo_7_O_24_·4H_2_O (99%), were purchased from the Sinopharm Chemical Reagent Co. Ltd. (Shanghai, China) They were used, as received, without further purification. Deionized water was used for all preparations.

### 2.2. Preparation of ZnAl-LDH Carriers

#### 2.2.1. The TP Process

Sodium hydroxide solution (20%) was added drop-wise to a vigorously stirred, mixed solution, containing the Zn/Al mixed solution (1:1), at constant pH of 9.5–10. After the resulting slurry was aged at 80 °C, for 18 h, the wet cake was thoroughly filtered and rinsed with deionized water. Finally, the sample was dried at 70 °C, for 12 h, and gently ground into the LDH-TP.

#### 2.2.2. The HS Process

Different from the TP, the sodium hydroxide solution and the Zn/Al mixed solution (1:1) were blended rapidly, in the process of HS. Afterwards, the formed slurry was transferred into stainless Teflon-lined 100 mL capacity autoclaves, then sealed and maintained at 120 °C, for 24 h, in an oven. After the hydrothermal treatment, the autoclaves were cooled down to the room temperature, filtered, washed thoroughly with deionized water, dried, and ground gently into the LDH-HS.

#### 2.2.3. The SG Process

Citric acid (2.5 mol/L) used as a binder was added into the Zn/Al mixed solution (1:1), through a peristaltic pump. Subsequently, an opaque faint yellow solution was finally observed, as the obtained mixed solution was stirred in 80 °C water bath, for 10 h, to evaporate the solvent. Moreover, the final sol was dried at 120 °C for 48 h, and ground gently into the LDH-SG.

#### 2.2.4. The UC Process

A solution of the Zn/Al mixed solution (1:1) was mixed with a urea solution (7.5 mol/L) in a 1000 mL round bottom flask, put into an oil bath at 120 °C; stirred and reflowed for 10 h. Then, the final slurry was filtered, washed, dried, and ground gently into the LDH-UC.

### 2.3. Mo/ZnAl-LDH Catalysts Preparation

The ZnAl-LDH support was impregnated in a 20 mL aqueous solution, containing 0.28 mol/L ammonium heptamolybdate. The pH of the mixture was maintained at 8 and the solution was maintained at room temperature for 12 h. After that, the resulting product was dried at 80 °C for 10 h and calcined at 400 °C for 1 h. The resulting solid prepared using the four kinds of preparation methods of support, were marked as Mo-TP, Mo-HS, Mo-SG and Mo-UC, respectively.

### 2.4. Characterization

The N_2_ adsorption-desorption on the surface of the Mo/ZnAl-LDH catalysts was measured on the Quantachrome NOVA-2000E apparatus (Quantachrome, FL, USA). By analysis of the adsorption-desorption isotherm of nitrogen, through the Brunauer-Emmett-Teller (BET) approach, the specific surface area of the samples was obtained. The pore size and volume distribution were calculated by the Brunauer-Joyner-Halenda (BJH) method. Before obtaining the adsorption-desorption isotherms, all catalysts were outgassed at a temperature of 150 °C.

X-ray diffraction (XRD) patterns of the catalyst was obtained on a Philips Panalytical X’Pert PRO X-ray diffractometer (PANalytical, Almelo, The Netherlands), using Cu Kα radiation, at a scan rate of 0.022θ s^−1^ in the 2θ range of 5°–80° to determine the phase structures of the obtained samples. The applied current and accelerating voltage were 30 mA and 40 kV, respectively. The Joint Committee of Powder Diffraction Standard (JCPDS) cards were used to manually analyze the diffraction patterns.

Zeta-potentials were measured using a zeta-potential analyzer (Brookhaven ZetaPALS, New York, NY, USA) and each value was reported as an average of five measurements.

X-ray photoelectron spectroscopy (XPS) was recorded on a Thermo Fisher Scientific ESCALAB 250 X-ray spectrometer (Waltham, MA, USA), fitted with a monochromatic Al excitation source (photon energy was at 1486.6 eV), operating at a transmitter current of 10 mA and a voltage of 15 kV. All spectra were calibrated to the C 1 s peak (binding energy = 284.6 eV) for the adventitious carbon, and all the datum of XPS were analyzed, using the Casa 2273 XPS software (Casa XPS 2.3.15, Quayside bookshop, Teignmouth, UK).

The morphology and particle size of the catalysts were characterized by Transmission electron microscopy (JEM-2100, JEOL, Tokyo, Japan), operated with an accelerating voltage of 200 kV, and magnification ranges of 0.5–1.5 million times. All the sample specimens for TEM studies were ultrasonically dispersed in ethanol, and one or two drops of the upper layer solution was pipetted out on a micro-mesh cupper grid. Before the TEM analysis, the sample specimens were vacuum dried in the microscope column, for 7–8 min.

### 2.5. Catalytic Activity Evaluation

In the CWAO process, cationic orchid X-BL was the representative of the organic wastewater. Figure 1 shows the structure of the cationic orchid X-BL. The degradation of the cationic orchid X-BL by the Mo/ZnAl-LDH catalyst was evaluated, using a simple beaker device, under room temperature and pressure. A sample of 1 g/L Mo/ZnAl-LDH catalyst was used to degrade 100 mg/L cationic orchid X-BL, with an air flux of 3.5 L/min. The pH of the simulated industry wastewater was not adjusted. During the CWAO process, the decomposition rate of cationic orchid X-BL was estimated on the basis of the absorbency, by UV-vis spectrophotometer (Shimadzu UV 2550, Kyoto, Japan) and Total Organic Carbon (TOC) analyzer (Shimadzu CPHCN200, Kyoto, Japan), to evaluate the mineralization of the cationic orchid X-BL, after reaction.

## 3. Results and Discussion

### 3.1. Crystal Structure of Mo/ZnAl-LDH Catalysts

The crystal phases of all as-prepared Mo/ZnAl-LDH catalyst are shown in Figure 2. All catalyst exhibited the diffraction lines arising from the Na_2_MoO_4_ phase (JCPDS 12-0773), located at 2θ = 16.95°, 27.76°, 35.66°, 48.98°, 52.18°, and 57.27°. Mo-SG and Mo-UC have extremely sharp and narrow diffraction spectra and exhibit high characteristic peaks of Na_2_MoO_4_, which indicates that Mo-SG and Mo-UC contain a single substance. However, Mo-TP and Mo-HS have extremely short and wide diffraction spectra and exhibit low characteristic peaks of Na_2_Mo_2_O_7_ (JCPDS 22-0906) and MoO_3_ (JCPDS 47-1320). This may be related to the phase transition because of its the long age time [21].

### 3.2. TEM Characterizations of the Mo/ZnAl-LDH Catalysts

The morphology of Mo/ZnAl-LDH catalysts characterized by TEM are exhibited in Figure 3. The lattice lines of Mo-TP and Mo-HS, with a smaller spacing of ~0.4 nm, showed heterogeneous particle size, morphology, some groups of dark thread-like fringes, wormlike appearance, and significant order of pore arrangement, consisting of the XRD characterization. As shown in Figure 3C,D, Mo-SG and Mo-UC, with a smaller spacing of ~0.6 nm, structural regularization, and single crystal structure, consists of the XRD characterization. For the particle of Mo-SG, the pore structure was the exclusive morphology, while the pore diameter was significantly larger than that of Mo-TP. The plentiful appearance difference may indicate the diverse Mo species and the variance of the specific surface area.

### 3.3. Textural Properties of Mo/ZnAl-LDH Catalysts

The adsorption-desorption isotherms of Mo/ZnAl-LDH catalyst are shown in Figure 4. All of the Mo/ZnAl-LDH catalysts exhibited the identical typical type IV adsorption-desorption isotherms with hysteresis loops, indicating that all catalysts showed mesoporous characteristics. BET surface area, pore size, and pore volume of all catalysts are listed in Table 1. The specific surface area of Mo-TP and Mo-HS was 2.8 m^2^/g and 3.9 m^2^/g, respectively, which were almost half of the Mo-UC, which was 5.8 m^2^/g. The specific surface area of Mo-SG was 15.4 m^2^/g, which was nearly five times as large as that of the Mo-TP. Zeta potential was the electrical potential which could further carry the adsorption behavior of the catalysts. All catalysts showed a negative zeta potential. For example, Mo-TP and Mo-HS were −14.4 mV and −18.4 mV, which were beneficial for the absorption of the cationic orchid X-BL, which is a kind of cationic azo dye with a positive Zeta potential.

### 3.4. XPS Characterization of the Mo/ZnAl-LDH Catalysts

XPS is an efficient method to investigate the surface composition and chemical state of the elements on the surface of as-prepared catalysts. Figure 5 shows the full XPS survey spectra of the Mo/ZnAl-LDH catalysts. Chemical composition of the Mo/ZnAl-LDH catalysts, with supports prepared using different methods, are shown in Table 2. The peaks of Mo, Zn, Al, and O were observed in all as-prepared catalysts. The element of zinc in the Mo-SG showed little change, as Zn 2p core-level spectra was almost completely transformed into the Zn-LMM. The percentage of Mo dosing in the Mo-TP and Mo-HS, were higher than that of the Mo-SG and Mo-UC, which meant that the TP and HS method promoted interactions between the Mo and the LDHs.

All Mo 3d XPS spectra exhibited the characteristic 3d 5/2 and 3d 3/2 doublet, caused by spin-orbit coupling of the Mo 3d orbitals (Figure 6). For the Mo-SG and the Mo-UC, a single assumption of a relative intensity ratio of the Mo 3d 5/2 and Mo 3d 3/2 lines, at 232.2 eV and 235.4 eV, was fixed to 3:2 which revealed two peaks for each Mo 3d 5/2 and Mo 3d 3/2 spin-orbit coupling [22,23]. For the Mo-TP and Mo-HS, the characteristic binding energy separation (Δ Mo 3d) between the Mo 3d 5/2 and 3d 3/2 was ∼3.1 eV, as observed previously in the literature [24]. The tiny chemical shift of the Mo 3d 5/2 at 231.8 eV and Mo 3d 3/2 at 234.9 eV implied that a tiny difference of the chemical composition and content occurred in the Mo/ZnAl-LDH catalysts. The Mo-TP and Mo-HS had a lower energy binding of Mo that was easier to transform into other valence states.

More information can be obtained from the comparison of the O 1s XPS spectra (Figure 7). Notably, both adsorption oxygen with a high binding energy and lattice oxygen with low binding energy, existed on the surface of the catalysts. The peak O 1s of the 529–531 eV corresponded to the O^2−^ ions, in the metal oxide, and the binding energy in the region of 531–533 eV was associated with surface adsorbed oxygen, from either surface hydroxyl or carbonate species. The peak after 532 eV (BE) was associated with adsorbed molecular water. The evidence was that the amount of O^2−^ ions in the Mo-TP and the Mo-HS was larger than in the Mo-SG and Mo-UC. The presence of surface hydroxyl or carbonate species played an important role in the selective oxidation process. However, large surface adsorbed oxygen could inhibit the catalytic activity.

### 3.5. CWAO Activity of the Mo/ZnAl-LDH Catalysts

The decolorization of the cationic orchid X-BL by the Mo/ZnAl-LDH catalysts was investigated in the CWAO process, under room temperature and pressure. In addition, mineralization efficiency of the cationic orchid X-BL was examined by the TOC analysis. As illustrated in Figure 8a,b, Mo-TP and Mo-HS showed better catalytic performance, compared with Mo-SG and Mo-UC. The superior decolorization ratio and the TOC removal efficiency of the cationic orchid X-BL were 84.2% and 73.9%, by Mo-TP, and 84.8% and 71.0%, by Mo-HS, respectively. To further verify the catalytic activity of the Mo/ZnAl-LDH catalysts, the adsorption efficiency was also measured for each sample (Figure 8c). The adsorption efficiency of all nano-hybrids was below 20%, suggesting that a catalytic reaction took place in the presence of air in the CWAO process. In general, the specific surface area was positive contact to the catalytic activity, as a catalyst with larger specific surface area could acquire more access with pollutants. Interestingly, the Mo-TP and Mo-HS catalysts possessed excellent catalytic activities with smaller specific surface area, as compared to the Mo-SG and Mo-UC. It was reasonable that Na_2_Mo_2_O_7_ was formed in the preparation of the Mo-TP and the Mo-HS. Na_2_Mo_2_O_7_ possessed an excellent catalytic activity and Na_2_MoO_4_ was not the active species, which was quite in line with the result [25,26,27]. In addition, O^2−^ ions in the oxygen deficient regions in Mo-TP and Mo-HS promoted the formation of active ·OH radicals [28]. The highest catalytic performance could be credited to appropriate crystalline from the XRD analysis, specific Mo species, lower zeta potential, higher active component of Mo and unique microgranular texture. Recycling was a parameter for evaluating the stability of the catalyst [29], and its results are show in Figure 8d. The discoloration rate by the Mo-TP and Mo-HS remained above 60%, after three reactions, however, the Mo-SG and Mo-UC were just under 20% of the discoloration rate, after two reactions. Therefore, the improvement of chemical stability for the Mo/ZnAl-LDH catalyst will be studied further in the next work.

## 4. Conclusion

The carrier of Zn-Al LDH with the four preparation methods have an influence on the catalytic performance of the Mo/ZnAl-LDH catalysts. The Mo-TP catalyst and the Mo-HS catalyst showed high catalytic performance for the degradation of cationic orchid X-BL, under room temperature and pressure. Na_2_Mo_2_O_7_ was detected in the Mo-TP and the Mo-HS, which contributed to the notably high catalytic activity in the organic wastewater degradation. The excellent performance could be attributed to the specific Mo species, low zeta position, high degree of Mo dispersion, and the unique texture properties.

## Figures and Tables

**Figure 1 materials-11-02390-f001:**
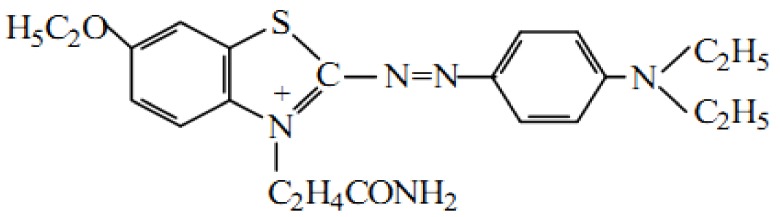
The structure of the cationic orchid X-BL.

**Figure 2 materials-11-02390-f002:**
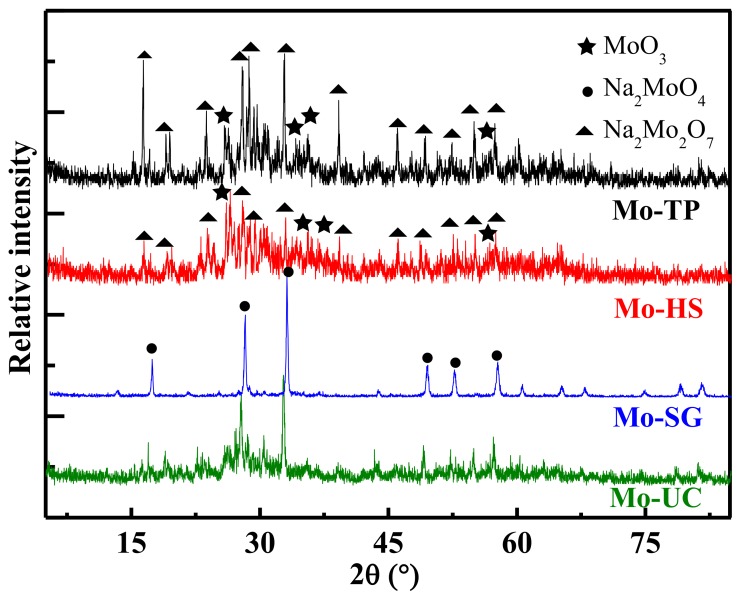
XRD patterns of the Mo/ZnAl-layered double hydroxide (LDH) catalysts.

**Figure 3 materials-11-02390-f003:**
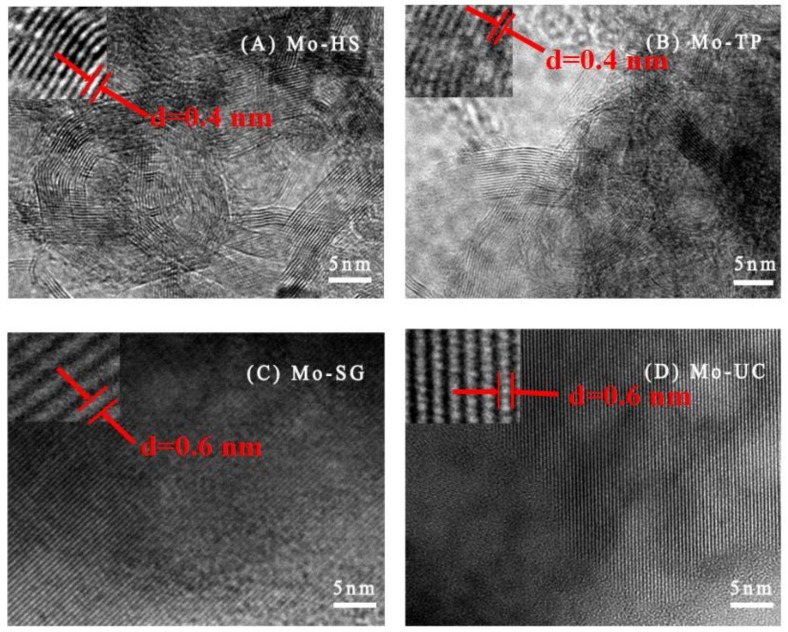
TEM images of Mo/ZnAl-LDH catalysts: (**A**) Mo-hydrothermal-synthesis (HS); (**B**) Mo-traditional-precipitation (TP); (**C**) Mo-sol-gel (SG); and (**D**) Mo-urea-co-precipitation (UC).

**Figure 4 materials-11-02390-f004:**
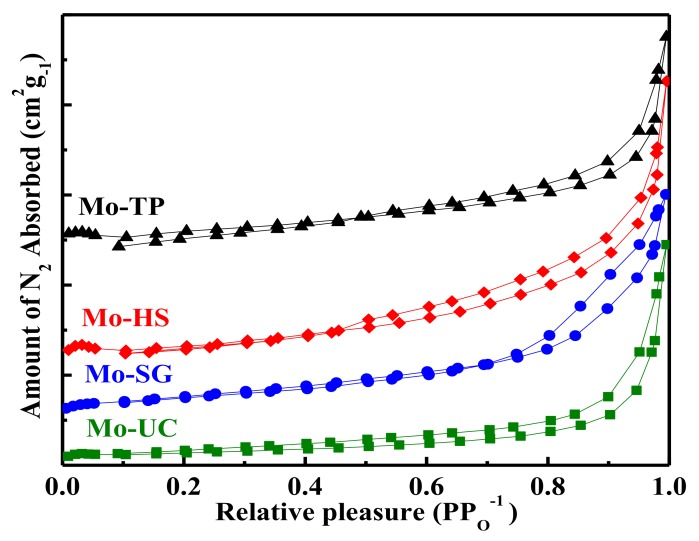
N_2_ adsorption-desorption isotherm curves of the Mo/ZnAl-LDH catalysts.

**Figure 5 materials-11-02390-f005:**
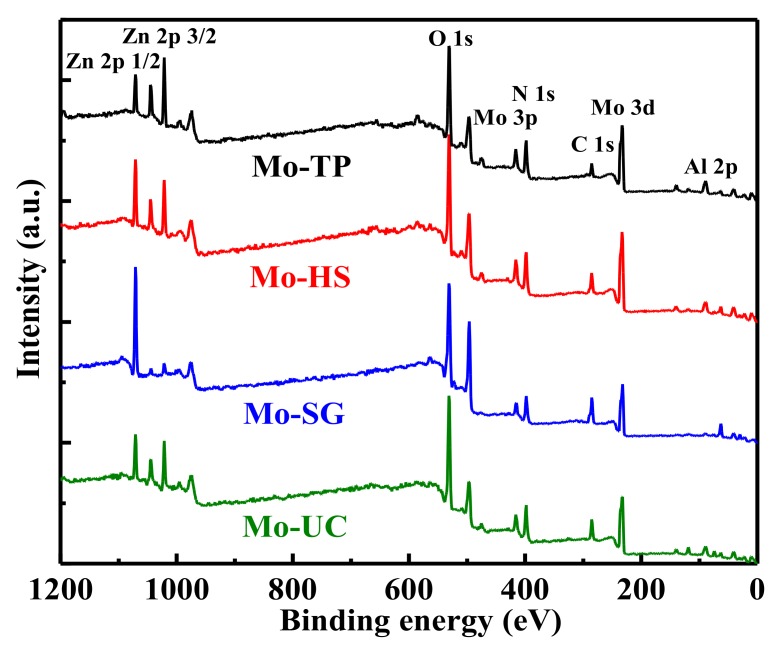
The X-ray photoelectron spectroscopy (XPS) survey spectra of the Mo/ZnAl-LDH catalysts.

**Figure 6 materials-11-02390-f006:**
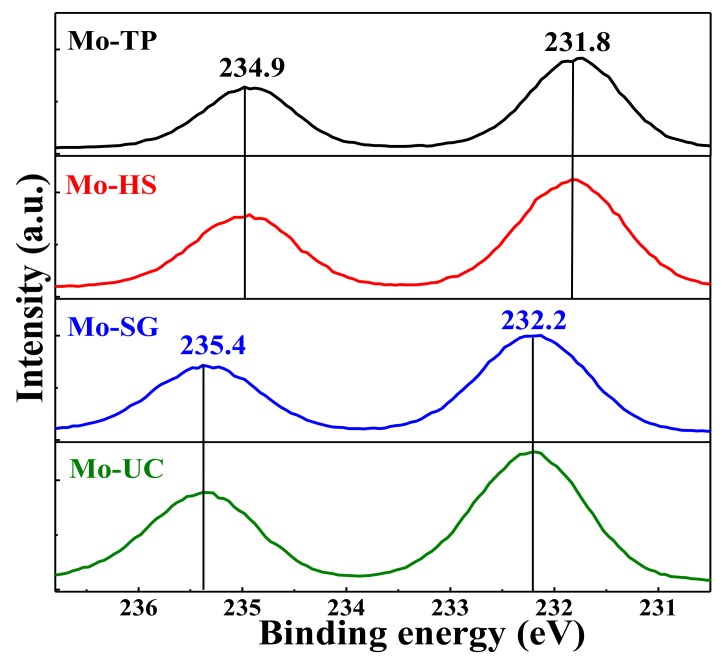
Mo 3d 3/2 and Mo 3d 5/2 peaks of the Mo/ZnAl-LDH catalysts.

**Figure 7 materials-11-02390-f007:**
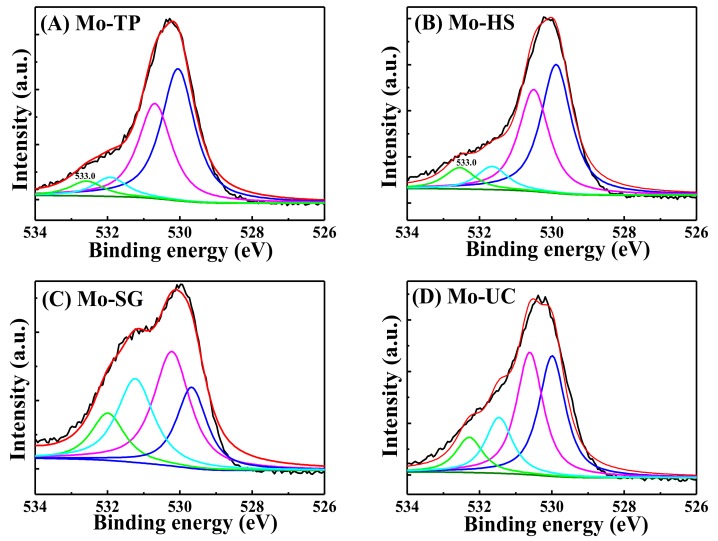
O 1s for the Mo/ZnAl-LDH catalysts: (**A**) Mo-traditional-precipitation (TP); (**B**) Mo-hydrothermal-synthesis (HS); (**C**) Mo-sol-gel (SG); and (**D**) Mo-urea-co-precipitation (UC).

**Figure 8 materials-11-02390-f008:**
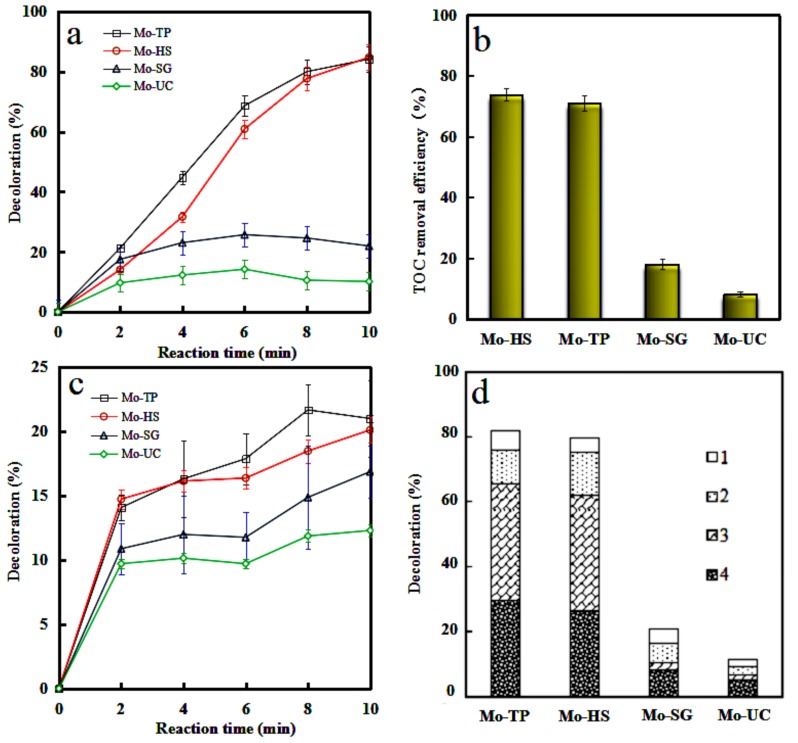
The degradation of Cationic Orchid X-BL (**a**,**b**) by the Mo/ZnAl-LDH catalysts in catalytic wet air oxidation (CWAO); the adsorption of Cationic Orchid X-BL (**c**) by the Mo/ZnAl-LDH catalysts; the recycling times (**d**) of the Mo/ZnAl-LDH catalysts.

**Table 1 materials-11-02390-t001:** The surface area, pore diameter, and pore volume of the Mo/ZnAl-LDH catalysts.

Catalysts	Specific Surface Area (m^2^/g)	Pore Volume (cm^3^/g)	Pore Diameter (nm)	Zeta Potential (mV)
Mo-TP	2.8	0.03	3.8	−14.4
Mo-HS	3.9	0.01	3.8	−18.4
Mo-SG	15.4	0.07	12.6	−20.2
Mo-UC	5.8	0.04	2.2	−11.2

**Table 2 materials-11-02390-t002:** Chemical composition obtained by the XPS spectra of the Mo/ZnAl-LDH catalysts.

Catalysts	Mo (at %)	Zn (at %)	Al (at %)	O (at %)	Na (at %)	C (at %)
Mo-TP	12.7	2.4	1.1	48.6	8.0	27.1
Mo-HS	15.0	3.5	0.7	55.8	7.4	17.7
Mo-SG	8.3	0.3	1.5	44.3	12.8	32.8
Mo-UC	10.3	2.0	6.7	23.4	5.4	52.2

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
