# Peer review of "Nanostructure Design and Catalytic Performance of Mo/ZnAl-LDH in Cationic Orchid X-BL Removal"

_materials, 2018, doi:10.3390/ma11122390_

Round 1
Reviewer 1 Report
Nanostructure Design and Catalytic Performance of Mo/ZnAl-LDH for Cationic Orchid X-Bl Removal
This manuscript studies ZnAl-LDH prepared with different tecnique: traditional precipitation, hydrothermal, sol-gel, urea co-precipitation and then impregnated with ammonium heptamolybdate with aim to obtained Mo-TP, Mo-HS, Mo-SG and Mo-UC. The resulting samples were contrasted through surface area, crystal structure, chemical state and morphology. Finally, the authors evaluated the degradation of cationic orchid X-BL under room temperature and pressure. The topic is interesting but there are some aspects that the authors should improve.
The manuscript could be published after revision.
Introduction:
The introduction should be revised. Since the paper is focused on degradation of cationic orchid X-BL, in the introduction should be added some references regarding specifically the issue of cationic orchid X-BL, underlining the novelty of the paper compering to the authors prevision paper: “High catalytic activity of Mo–Zn–Al–O catalyst for dye degradation: Effect of pH in the impregnation process” Applied Catalysis B: Environmental 160–161 (2014) 115–121.
Results:
I think it is important to add also the physical-chemical proprieties of the sample prepared support ZnAl-LDH.
The system Mo-TP and Mo-HS seems to have the same behavior in the catalytic removal of Cationic Orchid X-Bl Removal, why? And what is the different with Mo-SG and Mo-UC? The authors attributed the different trend to the Specific Surface Area, Crystal structure, but I think that the different trend could be attributed to Na2MoO4 and Na2Mo2O7 that is not present on Mo-SG and Mo-UC.
Other
Title: replace Nanostructure Design and Catalytic Performance of Mo/Znal-ldh for Cationic Orchid X-Bl Removal with Nanostructure Design and Catalytic Performance of Mo/ZnAl-LDH for Cationic Orchid X-Bl Removal
Check the spaces in the XPS text: for example replace Mo3d 5/2 with Mo 3d 5/2
Author Response
Response to Reviewer 1 Comments
This manuscript studies ZnAl-LDH prepared with different technique: traditional precipitation, hydrothermal, sol-gel, urea co-precipitation and then impregnated with ammonium heptamolybdate with aim to obtained Mo-TP, Mo-HS, Mo-SG and Mo-UC. The resulting samples were contrasted through surface area, crystal structure, chemical state and morphology. Finally, the authors evaluated the degradation of cationic orchid X-BL under room temperature and pressure. The topic is interesting but there are some aspects that the authors should improve.
The manuscript could be published after revision.
Response: We highly appreciate your consideration for our works, and many valuable recommendations for our further improvements. We have checked the manuscript and revised it.
Point 1: The introduction should be revised. Since the paper is focused on degradation of cationic orchid X-BL, in the introduction should be added some references regarding specifically the issue of cationic orchid X-BL, underlining the novelty of the paper compering to the authors prevision paper: “High catalytic activity of Mo-Zn-Al-O catalyst for dye degradation: Effect of pH in the impregnation process” Applied Catalysis B: Environmental 160-161 (2014) 115-121.
Response 1: Cationic orchid X-BL was representative as dye wastewater and some references regarding dye wastewater were added in the revised manuscript. This paper is focused on the effect of support ZnAl-LDH preparation method on the catalytic activity of Mo/ZnAl-LDH catalyst, which is difference with our previous paper.
Point 2: I think it is important to add also the physical-chemical proprieties of the sample prepared support ZnAl-LDH.
Response 2: XRD patterns of the sample prepared support ZnAl-LDH have been investigated. ZnAl-LDH prepared with TP and UC method has the diffraction of LDH, LDH-SG exhibited amorphous state and metal oxide structure was present in XRD pattern of LDH-HS. However, the structure of catalyst has different from support LDH because of impregnation and calcination process, thus the physical-chemical proprieties of the sample prepared support ZnAl-LDH have been not listed in the manuscript.
Point 3: The system Mo-TP and Mo-HS seems to have the same behavior in the catalytic removal of Cationic Orchid X-Bl Removal, why? And what is the different with Mo-SG and Mo-UC? The authors attributed the different trend to the Specific Surface Area, Crystal structure, but I think that the different trend could be attributed to Na2MoO4 and Na2Mo2O7 that is not present on Mo-SG and Mo-UC.
Response 3: Na2Mo2O7 was formed in the preparation of Mo-TP and Mo-HS, which was proved to possess an excellent catalytic activity. And Mo-TP and Mo-HS have a lower energy binding of Mo that is easier to transform into other valence state. So Mo-TP and Mo-HS seem to have the same behavior in the catalytic removal of cationic orchid X-BL removal. The Na2MoO4 was formed in the preparation of Mo-SG and Mo-UC, but it was not the active species. Thus the highest catalytic performance can be credited to Na2Mo2O7.
Point 4: Title: replace Nanostructure Design and Catalytic Performance of Mo/Znal-ldh for Cationic Orchid X-Bl Removal with Nanostructure Design and Catalytic Performance of Mo/ZnAl-LDH for Cationic Orchid X-Bl Removal.
Response 4: Thank you for your suggestion. In submitting process, the title has auto formed to “Mo/Znal-ldh”. The title “Nanostructure design and catalytic performance of Mo/ZnAl-LDH for cationic orchid X-BL removal” is original in the manuscript.
Point 5: Check the spaces in the XPS text: for example replace Mo3d 5/2 with Mo 3d 5/2.
Response 5: Thank you for your suggestion, we have checked and revised it in line 202, 204, 205, 206, 207, 208, 209, 213, 214, 216 and 220 and Fig.5 of the manuscript.

Reviewer 2 Report
The manuscript describes Catalytic Performance of 2 Mo/Znal-Ldh for Cationic Orchid X-Bl Removal. The composite catalysts preparation, characterizations and applications are interesting and important; but, the manuscript is suffering critical scientific presentation and results interpretations. The manuscript does not present the novelty of the work clearly in the aspects of the preparation and characterizations of the introduced composite. The introduction section needs to be revised to address the below mentioned comments. The authors should solve the below critical mentioned drawbacks and resubmit it to the journal. The existed critical drawbacks, unclear results interpretation and deficiencies make it very incomprehensible for publication.
Some weaknesses and disadvantages of the presented protocol are listed and highlighted below for the editor and author considerations.
The manuscript should be resented precisely and professionally. The authors should explain and justify the process preparation by low temperature procedure. They should clearly explain the process steps, investigate each effects and verify the process proof by scientific experimental results.
The authors should include further evidences and experimental results such as TEM-EDX map, or FESEM-EDX map.
The quality of the images and figures is not good enough for publication. The authors should improve the quality of the TEM scale bars. They should rewrite the scales clearly.
The author should eliminate the current grammatical, spelling, verb tense singular and plural, punctuation mark errors (commas, italics and so on) and also should confirm the correct scientific English. It is suggested to avoid or insert commas on the right positions. The English of the manuscript needs improvement. The authors should work and elevate the scientific English of the manuscript.
The authors should mention the complete terms of abbreviations before the parenthesis for the first time use when an abbreviation is used on the abstract and main manuscript.
The authors should revise the introduction section of the manuscript to clarify and justify the importance of the project. Otherwise, there is no advantage or novelty introduced on this project.
The authors should explore/include the previously similar published articles and compare them with their strategy on the introduction of the revised manuscript.
The abstract should be revised to clearly show the novelty of the work.
The authors should include the brand and model of all applied instruments on the revised manuscript. They should also comprehensively explain the utilized methods and techniques.
The article should be rewritten scientifically and professionally in order to present the work properly and being published.
There are important composite catalyst recycling references need to be cited on the revised manuscript. The below references should be cited in the revised manuscript:
Applied Catalysis B: Environmental, Volumes 160–161, 2014, 115-121.
Green Chem. 20, 3809-3817.
The title of the manuscript should be corrected, The spellings and capital abbreviations including “Al” should be corrected?
The preparation process conditions should be explained in detail including the method of the products preparation conditions and detail characterizations.
The recycling section of the manuscript should be developed and explained completely in the revised manuscript.
The Y axis of Fig. 4 does not have N2 numbers. The authors should revise all figures to include the related labels.
Author Response
The manuscript describes Catalytic Performance of 2 Mo/Znal-Ldh for Cationic Orchid X-Bl Removal. The composite catalysts preparation, characterizations and applications are interesting and important; but, the manuscript is suffering critical scientific presentation and results interpretations. The manuscript does not present the novelty of the work clearly in the aspects of the preparation and characterizations of the introduced composite. The introduction section needs to be revised to address the below mentioned comments. The authors should solve the below critical mentioned drawbacks and resubmit it to the journal. The existed critical drawbacks, unclear results interpretation and deficiencies make it very incomprehensible for publication. Some weaknesses and disadvantages of the presented protocol are listed and highlighted below for the editor and author considerations.
Response: We highly appreciate your consideration for our works, and many valuable recommendations for our further improvements. We have checked the manuscript and revised it.
Point 1: The manuscript should be resented precisely and professionally. The authors should explain and justify the process preparation by low temperature procedure. They should clearly explain the process steps, investigate each effect and verify the process proof by scientific experimental results.
Response 1: Thank you for your suggestion. The manuscript has been revised precisely and professionally. The process preparation has been clearly introduced in Materials and Methods of the manuscript. For example, TP process: Sodium hydroxide solution (20%) was added drop-wise to a vigorously stirred mixed solution containing the Zn/Al mixed solution (1:1) at constant pH of 9.5–10. After the resulting slurry was aged at 80 ℃ for 18 h, the wet cake was thoroughly filtered and rinsed with deionized water. Finally the sample was dryed at 70 ℃ for 12 h, and ground gently into LDH-TP.
The experimental results have been discussed carefully in the revised manuscript.
Point 2: The authors should include further evidences and experimental results such as TEM-EDX map, or FESEM-EDX map.
Response 2: Thanks. SEM-EDS has been measured and the results was consist with the chemical composition obtained by XPS spectra of Mo/ZnAl-LDH catalysts. We have found that the highest catalytic performance can be credited to Na2Mo2O7 from XRD pattern and XPS spectra. The chemical element distribution by TEM-EDX map or FESEM-EDX map will be investigated in our future work.
Point 3: The quality of the images and figures is not good enough for publication. The authors should improve the quality of the TEM scale bars. They should rewrite the scales clearly.
Response 3: As suggested, we have already improved the quality of the images and figures such as Fig. 7 and Fig. 8. TEM scale bars have been revised and also the distance has been added in Fig. 3.
Point 4: The author should eliminate the current grammatical, spelling, verb tense singular and plural, punctuation mark errors (commas, italics and so on) and also should confirm the correct scientific English. It is suggested to avoid or insert commas on the right positions. The English of the manuscript needs improvement. The authors should work and elevate the scientific English of the manuscript.
Response 4: We have checked grammatical, spelling, verb tense singular and plural, punctuation mark errors carefully and revised in the manuscript such as in line 28, 30, 47, 57, 61, 62, 66, 83, 89, 90, 94, 105 and 112.
Point 5: The authors should mention the complete terms of abbreviations before the parenthesis for the first time use when an abbreviation is used on the abstract and main manuscript.
Response 5: Thank you for your suggestion, we have revised it carefully. For example, in line 11, 12, 13, 15 and 16 of the manuscript.
Point 6: The authors should revise the introduction section of the manuscript to clarify and justify the importance of the project. Otherwise, there is no advantage or novelty introduced on this project.
Response 6: In the introduction, we have added the disadvantage of CWAO which need high temperature and high pressure. In this manuscript, the novelty is that catalyst has used in CWAO process in the degradation of dye wastewater under room temperature and pressure.
Point 7: The authors should explore/include the previously similar published articles and compare them with their strategy on the introduction of the revised manuscript.
Response 7: We have added the previously similar published articles and revised it carefully. The effect of support ZnAl-LDH preparation method on the catalytic activity of Mo/ZnAl-LDH catalyst was investigated in this manuscript, which is difference with our previous paper.
Point 8: The abstract should be revised to clearly show the novelty of the work.
Response 8: Thank you for your suggestion, the novelty of the work is that catalyst has used in CWAO process in the degradation of dye wastewater under room temperature and pressure. In addition, LDHs preparation methods make a difference in the performance of Mo/ZnAl-LDH.
Point 9: The authors should include the brand and model of all applied instruments on the revised manuscript. They should also comprehensively explain the utilized methods and techniques.
Response 9: As suggested, we have added the brand and model of all applied instruments in the revised manuscript such as in line 119 and 127.
Point 10: The article should be rewritten scientifically and professionally in order to present the work properly and being published.
Response 10: Thank you for your suggestion, we have checked and revised it carefully.
Point 11: There are important composite catalyst recycling references need to be cited on the revised manuscript. The below references should be cited in the revised manuscript:
Applied Catalysis B: Environmental, Volumes 160–161, 2014, 115-121.
Green Chem. 20, 3809-3817.
Response 11: Thank you for your suggestion, more references related our present work have been cited in the revised manuscript.
Point 12: The title of the manuscript should be corrected, The spellings and capital abbreviations including “Al” should be corrected?
Response 12: In submitting process, the title has auto formed to “Mo/Znal-ldh”. The title “Nanostructure design and catalytic performance of Mo/ZnAl-LDH for cationic orchid X-BL removal” is original in the manuscript.
Point 13: The preparation process conditions should be explained in detail including the method of the products preparation conditions and detail characterizations.
Response 13: Thanks, the preparation process conditions has been clearly introduced in Materials and Methods of the manuscript. For example, Mo/ZnAl-LDH catalysts preparation: the ZnAl-LDH support was impregnated in a 20 ml aqueous solution containing 0.28 mol/L ammonium heptamolybdate. The pH of the mixture was maintained at 8 and the solution was maintained at room temperature for 12 h. After that, the resulting product was dried at 80 ℃ for 10 h and calcined at 400 ℃ for 1 h. The resulting solid prepared using four kinds of preparation method of supports were marked as Mo-TP, Mo-HS, Mo-SG and Mo-UC, respectively.
Point 14: The recycling section of the manuscript should be developed and explained completely in the revised manuscript.
Response 14: Thank you for your suggestion, we have added the recycling section of the manuscript such as in line 250, 251, and 252.
Point 15: The Y axis of Fig. 4 does not have N2 numbers. The authors should revise all figures to include the related labels.
Response 15: As suggested, In order to separate the four curves in an origin figure, the starting point of the Y axis of numbers is different. So some figures are unmarked the Y axis of numbers.

Round 2
Reviewer 1 Report
Accept in present form
Reviewer 2 Report
The revised manuscript has been improved significantly compared to the first submitted one. The authors have included the referees' comments and corrected the short comes of the first submitted manuscript.
The below corrections must be done before the manuscript be published.
- The complete terms of the abbreviations in the introduction section must be written as follows:
Do not delete the complete terms of the abbreviations denoted for the first time use. Please correct the below abbreviations as below:
P. 1, Line 27, "Catalytic wet air oxidation (CWAO)"
P. 1, Line 40, "Layer double hydroxides (LDHs)"
P. 1, Line 43, "urea co-precipitation (UC)"
P.1. Line 43, "sol-gel (SG)"
P. 1, Line 43, "traditional precipitation (TP)"
P. 1, Line 44, "hydrothermal synthesis (HS)"
After these minor corrections the manuscript is ready for publication. I suggest this manuscript will be accepted for the publication after approve of the editor.